# Replacing Mineral Fertilisers for Bio-Based Fertilisers in Potato Growing on Sandy Soil: A Case Study

Chantal M. J. Hendriks [1,*,†], Vaibhav Shrivastava [2,†], Ivona Sigurnjak [2], Jan Peter Lesschen [1], Erik Meers [2], Rembert van Noort [3], Zhongchen Yang [4] and Rene P. J. J. Rietra [1]

1   Team Sustainable Soil Management, Wageningen Environmental Research, Wageningen University & Research, P.O. Box 47, 6700 AA Wageningen, The Netherlands; janpeter.lesschen@wur.nl (J.P.L.); rene.rietra@wur.nl (R.P.J.J.R.)
2   Department of Green Chemistry and Technology, Faculty of Bioscience Engineering, Ghent University, Coupure Links 653, 9000 Ghent, Belgium; Vaibhav.Shrivastava@UGent.be (V.S.); Ivona.Sigurnjak@UGent.be (I.S.); Erik.Meers@UGent.be (E.M.)
3   Southern Agriculture and Horticulture Organisation (ZLTO), Onderwijsboulevard 225, 5223 DE 's-Hertogenbosch, The Netherlands; rembert@scarabaeus.nl
4   Team Animal Drug 2, Wageningen Food Safety Research, Wageningen University & Research, P.O. Box 47, 6700 AA Wageningen, The Netherlands; zhongchen.yang@wur.nl
*   Correspondence: chantal.hendriks@wur.nl
†   These authors contributed equally to this manuscript.

**Featured Application: Use bio-based fertilisers as a replacement for mineral fertiliser or slurry manure.**

**Abstract:** The refinement level of bio-based fertilisers (BBFs) can influence environmental and agronomic performance. This study analyses the environmental and agronomic effect of different BBFs on potato growing in sandy soil. A less refined product (liquid fraction of digestate (LFD)), two refined products (ammonium sulphate (AS) and potassium concentrate (KC)), and mineral fertilizer (MF) are compared by conducting: (i) a nitrogen (N) incubation experiment where the N release rate of the BBFs is determined, (ii) a greenhouse gas emission experiment where $N_2O$, $CO_2$, and $CH_4$ emissions after BBF application are measured, (iii) a pot experiment where the nutrient fertiliser replacement value (NFRV) of the BBF is calculated, and (iv) a full-scale field trial where the potato quality and quantity and the remaining N residues in the soil after harvest are assessed. The N release rate and the NFRV of AS (142 ± 19% and 1.13, respectively) was higher compared with the LFD (113 ± 24% and 1.04) and MF (105 ± 16% and 1.00). Lowest $N_2O$ emissions were observed after the application of the less refined product (0.02 ± 0.01 per 100 g N applied) and highest for MF urea (0.11 ± 0.02 per 100 g N applied). In the full-scale field trial, no significant difference in potato yield was observed in the plots that received manure in combination with BBF or MF. This study showed that all three BBFs can safely be used in potato growing on sandy soils. However, the adoption of BBFs can be stimulated by (i) solving the practical issues that occurred during the application of LFD, (ii) making sure BBFs are on the list of RENURE materials so they can legally replace mineral fertiliser, and (iii) reducing the surplus of slurry manure to stimulate the use and fair pricing of BBF products.

**Keywords:** agricultural circularity; sustainable agriculture; environmental impact; manure processing; GHG emissions; fertiliser replacement value

## 1. Introduction

European farming systems are among the most efficient production systems in the world [1]. However, these systems are often input-intensive as they consume large quantities of mineral fertiliser, water, fuel and pesticides [2]. In Europe, 46% of the total N applied to agricultural soil comes from mineral fertiliser [3]. A transition towards sustainable

agriculture is needed, not only to reduce the environmental impact of agricultural systems, but also to become more resilient to economic and societal challenges such as fluctuating production prices, changing consumer behaviour, and extreme weather events. Awareness of the urgency for sustainable and resilient farming systems has increased among policy makers, businesses, and the scientific community [4]. The European Commission emphasises this by the large number of policy initiatives that were recently established: the EU Green Deal, Farm to Fork Strategy, Chemicals Strategy for Sustainability, New EU Strategy on Adaptation to Climate Change, Organic Farming Action Plan, Zero Pollution Action Plan for Air, Water and Soil, New Soil Strategy, Fertilizing Product Regulation revision, and the Fit for 55 Climate Package.

One way to stimulate the transition towards sustainable and resilient farming systems is circular agriculture because it aims to minimise external inputs and negative discharges to the environment, and to close nutrient cycles. Therefore, it allows crop and livestock production without the depletion of non-renewable sources and harming the environment. In 2018, the European agricultural sector consumed 10.2 million tonnes of mineral nitrogen (N) fertiliser and 1.1 million tonnes of mineral phosphorus (P) fertiliser [5]. Reducing these numbers by using technologies that valorise biowaste into bio-based fertiliser (BBF) products can stimulate circularity [6–10]. Different techniques have been developed for the recycling of nutrients from biowaste [11]. For example, liquid and solid separation can be carried out through centrifuge, screw or belt press, and pressurized membrane filtration techniques such as microfiltration, ultrafiltration, or reversed osmosis can be used to refine the products [11,12]. BBF products that meet the criteria of being referred to as recovered nitrogen from manure (RENURE) are safe to use as N fertiliser [13]. However, criteria define quality and handling rules that BBF should comply in order to be classified as RENURE [13]. This process of being classified as RENURE product is still ongoing for several BBFs.

The adoption of (potential) RENURE products in the agricultural sector depends on collaborations between the biowaste producers (e.g., pig farmers), biowaste processing industry, and end users of BBFs. This study collaborated with all three stakeholders and in combination with laboratory and pot and field experiments, this study provides a unique and complete picture of the potential adoption of the tested BBFs. This way of analysing the adoption of BBFs is important because each stakeholder has different interests [6]. For example, factors influencing the decision of end users to adopt BBFs depends especially on the agronomic efficiency of BBFs [14]. These end users are questioning whether the agronomic efficiency of refined or less refined BBFs differs. Therefore, the hypothesis of this study is to test whether refined BBFs perform environmentally and agronomically better compared with less-refined BBFs. This study aims to analyse the environmental and agronomic effect of using BBFs that differ in refinement level.

## 2. Materials and Methods

The following experiments were carried out in this study: (i) a nitrogen (N) incubation experiment where the N release rate of the BBFs is determined, (ii) a greenhouse gas emission experiment where the $N_2O$, $CO_2$, and $CH_4$ emissions after BBF application are measured, (iii) a pot experiment where the nutrient fertiliser replacement value (NFRV) of the BBF is calculated, and (iv) a full-scale field trial where the potato quality and quantity and the remaining N residues in the soil after harvest are assessed.

### 2.1. Study Area

2.1.1. Bio-Based Fertiliser Products

Three BBFs were investigated in this study and compared with mineral fertiliser (MF): liquid fraction of digestate (LFD); potassium concentrate (KC); and ammonium sulphate solution (AS). The BBFs were obtained from a mesophilic (38 °C) anaerobic co-digestion (AD) plant located at the premises of a pig farmer in Oirschot (the Netherlands). The biogas installation produces 1000 $m^3$ gas $h^{-1}$, which corresponds to 9.8 MWh. The AD plant has a

hydraulic retention time of 30 days, and an input feed consisting of 60% pig slurry and 40% plant-based products (i.e., crop residues, food waste). LFD was obtained after mechanical separation of raw digestate by means of a belt filter press system. The product was further refined for the production of AS and KC by inducing ammonia volatilisation from LFD in a 4-stage thermal vacuum evaporation system during increased temperature regime. The ammonia rich gas stream was washed by $H_2SO_4$ solution to dissolve ammonia and form AS. After the evaporation of LFD, most of the N was removed and the remaining product is characterised by a high concentration of suspended solids and potassium, referred to as KC.

The chemical composition of the BBFs is reported in Table 1. Each BBF was analysed by a certified laboratory that uses standardized methods, and therefore it was assumed that the analysis of a single sample was representative for the entire BBF batch. The same applies to the other analyses carried out in this study. The dry matter (DM) content was determined as the residual weight after 48 h of drying at 105 °C. Organic matter (OM) of the applied products was determined by ashing the samples at 550 °C for 5 h. Total N was determined using Kjeldahl destruction, whereas $NH_4^+$-N and $NO_3$-N (excluded from Table 1) were measured colourimetrically by a continuous flow auto-analyser (Chemlab System 4, Skalar, the Netherlands) after subsequent extraction in 1 M KCl, conform ISO 14256. The organic N was further calculated by subtracting mineral N (i.e., $NH_4^+$-N and $NO_3$-N) from total N. The total carbon was analysed by using dumas dry combustion method. The total organic carbon (TOC) was measured using the MachereyNagel 985093 method and measuring the solution in a P-PRO-32 spectrophotometer (Macherey-Nagel, Düren, Germany). To analyse the P and K content, the BBFs were pre-treated according to NEN7431, P and K were extracted according to NEN7433, and analysed according to NEN7435. During pre-treatment, the products were homogenised, dried, and crushed. For the extraction, sulphuric acid, hydrogen peroxide, and copper sulphate were used. The P content was analysed using the continuous flow auto-analyser (Chemlab System 4, Skalar, the Netherlands), and the K content was analysed using inductively coupled plasma optical emission spectrometry (ICP-OES) (Varian Vista MPX, Santa Clara, California, USA). The sulphur (S) content was measured by DIN EN ISO 11885: 2009-09, EG2003/2003. The pH and electrical conductivity (EC) were determined on the fresh sample using an Orion-520A pH meter (USA) and Orion-star A212 conductivity meter, respectively. All three experiments used the BBFs from the same batch to avoid potential differences in chemical composition of the tested products.

**Table 1.** The chemical composition of ammonium sulphate (AS), potassium concentrate (KC), liquid fraction of the digestate (LFD), and pig slurry manure (Man).

| | Dry Matter % | OM g kg⁻¹ | N g kg⁻¹ | NH₄⁺-N g kg⁻¹ | Organic N g kg⁻¹ | Total C g kg⁻¹ | TOC g L⁻¹ | C/N Ratio | P g kg⁻¹ | K g kg⁻¹ | S g kg⁻¹ | pH | EC mS cm⁻¹ |
|---|---|---|---|---|---|---|---|---|---|---|---|---|---|
| AS | 36 | N/A | 81.6 | 81.6 | N/A | 0.3 | 0.1 | N/A | 0.1 | N/A | 11.9 | 2.8 | 278 |
| KC | 12 | 69 | 6.4 | 2.5 | 3.9 | 41.4 | 27.8 | 6.5 | 4.9 | 20.7 | 3 | 7.4 | 59 |
| LFD | 3 | 9 | 4.5 | 3.7 | 0.8 | 9.2 | 2.8 | 2 | 0.7 | 6.9 | 0.1 | 8.0 | 44 |
| Man | | | 3.8 | | | | | | 2.4 | 4.0 | | | |

TOC: total organic carbon; EC: electrical conductivity; OM: organic matter; total K and OM were not detected in AS solution.

### 2.1.2. Bio-Based Fertiliser End User

The agronomic and environmental performance of two BBFs were tested in a crop field of a potato farmer that specialises in using precision agriculture. This field experiment was carried at a field in Eersel (51°21′14.6″ N, 5°19′58.0″ E), 34 km from the location where the BBFs were produced. The crop rotation is based on one year of potato and three years of maize. The soil is typically referred to as a plaggic Anthrosol [15], which indicates that the topsoil is enriched by OM through the plaggic farming system that existed in the area before mineral fertiliser started to be used. The subsoil is poor in OM and consists of cover sand, which was deposited during the Weichselian glaciation. The elevation of the field

ranges from 25.4 m above sea level in the east to 26.2 m above sea level in the west. The weather during the growing season was extremely dry and warm (Appendix A) and the farmer did not have the capacity to apply irrigation.

Soil of the topsoil layer (0–30 cm, loamy sand) was collected from this crop field in April 2020 and used for the experiments on the N release rate, the GHG emissions, and the nutrient fertiliser replacement value (NFRV). The soil characteristics are given in Table 2. Near infrared spectroscopy (NIRS) and $CaCl_2$ extraction, common practices of the laboratory Eurofins, were used for the soil analysis and the analysis was elaborated with a fertiliser recommended for consumption potato (Table 2). The soil of the potato field is rich in potassium (K) and therefore it is not recommended to test the NFRV of KC on this soil. A K-poor soil was used instead for NFRV determination (Wageningen, 51°59′14.8″ N, 5°39′54.4″ E). The texture of the K-poor soil is also loamy sand (88% sand, 6% silt, and 2% clay), with a pH-$H_2O$ of 6, OM content of 3.6%, an initial total N and mineral N content of 3930 kg N ha$^{-1}$ and 45 kg N ha$^{-1}$, respectively (Table 2). For the analyses, the N or K application rate varied while other nutrients were kept at optimal condition by following the fertiliser recommendation for consumption potatoes.

**Table 2.** Initial characteristics of the two soils that were used for this study and the fertiliser recommendations for growing consumption potatoes.

| Soil Characteristics | | | Fertiliser Recommendation | | |
|---|---|---|---|---|---|
| | Potato Field Soil | K-Poor Soil | | Potato Field Soil | K-Poor Soil |
| N total stock (kg ha$^{-1}$) | 4030 | 3930 | | | |
| N-supply capacity (kg ha$^{-1}$) | 60 | 45 | N kg ha$^{-1}$ yr$^{-1}$ | 310 | 310 |
| C/N-ratio | 13 | 17 | | | |
| Plant available S (kg ha$^{-1}$) | 4 | 161 | $SO_3$ kg ha$^{-1}$ yr$^{-1}$ | 23 | 23 |
| Plant available P (kg ha$^{-1}$) | 6.5 | 5.4 | $P_2O_5$ kg ha$^{-1}$ yr$^{-1}$ | 0 | 60 |
| Plant available K (kg ha$^{-1}$) | 500 | 115 | $K_2O$ kg ha$^{-1}$ yr$^{-1}$ | 70 | 265 |
| Plant available Ca (kg ha$^{-1}$) | 190 | 55 | CaO kg ha$^{-1}$ yr$^{-1}$ | 75 | 75 |
| Plant available Mg (kg ha$^{-1}$) | 300 | 115 | MgO kg ha$^{-1}$ yr$^{-1}$ | 0 | 21 |
| Plant available Na (kg ha$^{-1}$) | <20 | <20 | | | |
| pH | 5.8 | 6 | | | |
| C-organic (%) | 1.5 | 2 | | | |
| OM (%) | 3.1 | 3.6 | Effective OM kg ha$^{-1}$ yr$^{-1}$ | 1020 | 1095 |
| Clay (%) | 2 | 2 | | | |
| Silt (%) | 12 | 6 | | | |
| Sand (%) | 83 | 88 | | | |

OM: organic matter.

### 2.2. N Release Rate of Bio-Based Fertilisers

An incubation experiment was set-up to test the mineralisation rate of BBFs. Prior to the incubations, the soil was air-dried for 5 weeks and sieved through a 2 mm screen. After air-drying and sieving, the collected soil was pre-incubated at 35% water-filled pore space (WFPS) for a week. Following existing procedures [16,17], 271 g of pre-incubated soil was mixed with AS, LFD, or KC, and calcium ammonium nitrate (CAN; 30%) that was used as a reference for the experiment. Additionally, unfertilised soil was used as a control for all treatments. After mixing the soil and respective fertiliser thoroughly, the mixture was filled in cylindrical tubes (dimensions: 18cm height, 3.6cm diameter) and compacted to the bulk density of 1.4 Mg m$^{-3}$ (10 cm compacted height). Subsequently, the WFPS of the mixture was increased to 50% using distilled water. A total of 120 tubes (4 replicates × 5 treatments × 6 sampling moments) were incubated for 120 days. The fertilisers were added at the dose corresponding to 170 kg total N ha$^{-1}$ yr$^{-1}$, which is the maximum permissible limit of livestock manure in the nitrate vulnerable zones [18].

At every 20 day interval, destructive sampling was done during which 10 g of soil per tube was mixed with 50mL of 1M KCl, and the suspension was shaken end-over-end for 30 min, and subsequently filtered. The filtrate was measured colourimetrically by a continuous flow auto-analyser (Chemlab System 4, Skalar, the Netherlands) for the determination of ammonium ($NH_4^+$-N) and nitrate ($NO_3^-$-N). For the N incubations, the net N-release ($N_{rel,net}$, Equation (1)) and N-mineralisation rate ($N_{min,net}$, Equation(2)) were calculated. The net N release is the difference between mineral-N available in the fertilized soil and mineral-N available in the control ($N_{control}$) divided over the total N applied in the fertilised treatment (Equation (1)) [19].

$$N_{rel,net}(\%) = \frac{\left([NO_3^--N,treatment]-[NO_3^--N,control]\right)+\left([NH_4^+-N,treatment]-[NH_4^+-N,control]\right)}{N_{total}\ applied} \times 100 \tag{1}$$

The calculation of $N_{min,net}$ was carried out via subtracting net N release rate at a particular time ($N_{rel,net}(t)$) with the amount of mineral N already present in the products at the start of the experiment (($N_{rel,net}(t = 0)$) [20] (Equation (2)).

$$N_{min,net}(t; \%total\ N) = N_{rel,net}(t) - N_{rel,net}(t = 0) \tag{2}$$

A positive value in the above equation signifies the net mineralisation and a negative value indicates immobilisation.

### 2.3. GHG Emissions of the Bio-Based Fertilisers after Soil Application

A GHG experiment was conducted under controlled conditions in soil microcosms to study the outcome of fertiliser addition on soil respiration under controlled conditions. For this, the air-dried soil was pre-incubated at 40% WFPS for 7 days followed by application of fertilisers at 170 kg N/ha soil and incubation at 80% WPFS. Each microcosm was made using a 1L borosilicate glass bottle (height—23.6 cm, diameter—10.8 cm) fitted with a GL45-thread smart cap (model: SW45-2A). Each smart cap has two 2 mm threaded openings that can be closed with blind plugs or fitted with valves to serve as inlet and outlet ports for sampling.

At 80% WFPS, 568 g of pre-incubated soil mixed with fertilisers was moderately compacted in Duran bottles to attain the bulk density of 1.3 kg m$^{-3}$. The slightly lower bulk density (compared with incubations) was chosen in this case to maintain effective aerobic conditions inside the mesocosm. A total of 15 mesocosms (3 replicates × 3 BBFs; 2 replicates each of CAN, urea and unfertilized blank) were established for the study. One of the two threaded openings on the smart cap were left open allowing aerobic respiration. Over a period of 18 days, the GHG monitoring was carried out by using a photoacoustic infrared spectroscopy multi-gas analyser (Gasera 1; Turku, Finland) calibrated for measurements of $CO_2$, $CH_4$, and $N_2O$. The measurement was carried out on days 0, 1, 2, 4, 7, 9, 11, 14, 16, and 18. The analyser was connected to the mesocosm via two 1 m long Teflon tubes with 2 mm inner diameter in a closed circuit. During the measurement, gases were pumped out of the headspace (at flow rate of 800 mL/min), passed through the analyser, and then returned to the mesocosm in a closed loop. Gas concentrations in the headspace of the mesocosms were measured at 4, 8, 12, and 16 min after connecting the tubing to the mesocosm. During each 4 min time step, the analyser detected the change in concentration of the measured gases.

For the gaseous emissions, the fluxes of $CO_2$, $CH_4$, and $N_2O$, were calculated from concentration change over time, considering the volume of the headspace, the piping, and the area of the soil surface. The measurement of $NH_3$ was not considered due to unreliability of photoacoustic gas analysers in measuring ammonia gas. The conversion of gas concentrations (in ppm) to emission flux was carried out using ideal gas law (Equation (3)) [21]:

$$Flux_{area} = \frac{\Delta gas}{\Delta t} \times \frac{P \times M \times n}{R \times T} \times \frac{V}{A} \tag{3}$$

where Flux $_{area}$ is the elemental flux released as a gas, in $\mu g \ m^{-2} \ h^{-1}$ or $\mu g \ kg^{-1} \ h^{-1}$; $\Delta gas / \Delta t$ is the slope of the linear regression of gas concentration against time; P is the pressure in the cell (0.838 atm); M is the molar mass of the element (e.g., 14 for N); n is the number of atoms of the element in the gas (e.g., 2 N in $N_2O$); R is the ideal gas constant (0.08206 L atm $mol^{-1} \ K^{-1}$); T is the average atmospheric temperature (294 K); V is the total volume of the headspace, tubing, and analyser cell (0.623 L); and A is the surface area of the soil in the mesocosm (0.0069 $m^2$).

A linear interpolation between two measurement days was used to compute the cumulative flow for each gas. In all cumulative fertiliser emissions, the cumulative fluxes obtained with the soil control were deducted. The emission factors (EF%) for the gases were expressed as an amount of fertiliser applied (in kg $ha^{-1}$) and was calculated using Equation (4) [21]:

$$EF\% = \frac{\left[ \text{cum gas flux }_{(fertilizer)} - \text{ cum gas flux}_{(control)} \right]}{N \ (or \ C) \ applied} \times 100 \tag{4}$$

where EF% is the emission factor ($N_2O$-N, $CO_2$-C, or $CH_4$-C emitted as a % of fertiliser applied); cum gas emission$_{(fertiliser)}$ and cum gas emission$_{(control)}$ are cumulative emissions in kg N $ha^{-1}$; and N (or C) applied is the N (or C) application rate in kg N (or C) $ha^{-1}$.

### 2.4. Nutrient Fertiliser Replacement Value

The NFRV quantifies the ability of BBF to replace mineral fertiliser in terms of nutrient supply to a crop [22–24] (Figure 1). The yield response curve method and the nutrient recovery method are two commonly used methods to assess the NFRV. In this study, the response yield curve method was used. This method first fits a yield curve based on the crop yields that correspond, in our case, to different mineral N or K fertiliser application rates. Secondly, the yields obtained from the treatments that received BBFs are also plotted in the graph. The NFRV can now be calculated by Equation (5):

$$NFRV = \frac{Y_2 - Y_0}{Y_1 - Y_0} \tag{5}$$

where, $Y_0$ is the yield obtained from the yield curve when no mineral fertiliser is applied, $Y_1$ is the yield obtained from the yield curve when a fixed amount of mineral fertiliser is applied, and $Y_2$ is the yield obtained from the treatment that received BBF.

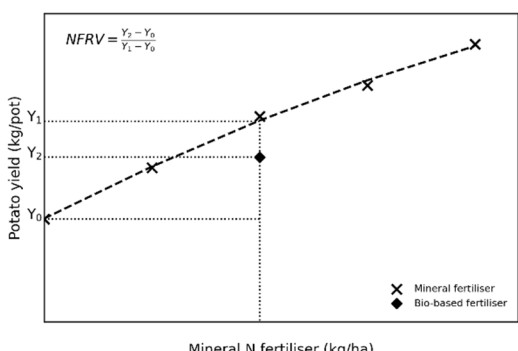

**Figure 1.** Calculation of the nitrogen fertiliser replacement value (NFRV) using the response yield curve method (adapted from [24]).

In a pot experiment with potato, the NFRV of the three BBFs was assessed. The fresh weight of the potato tubers was used to indicate the crop yield. A previous study mentioned that obtaining the NFRV based on the fresh or dry weight or the N uptake is valid although the value of NFRV of BBFs obtained by fresh or dry weight might be slightly higher [25]. To be able to draw a yield curve, 0%, 20%, 50%, 70%, and 100% of the

recommended amount of N or K was applied, which corresponds to 0 kg, 62 kg, 124 kg, 217 kg, and 310 kg N ha$^{-1}$, and for the K-treatments to 0 kg, 53 kg, 106 kg, 186 kg, and 265 kg K ha$^{-1}$ (Table 2). Mineral fertilisers CAN and KCl were used for these treatments as they are the most commonly used synthetic counterparts for the used BBFs. A deficit amount of BBF (40% of the recommended amount of N or K) was applied to test the difference in plant growth between plants that received mineral fertiliser and plants that received BBF. Other nutrients were kept at optimal conditions, which correspond to the fertilisation recommendation of consumption potatoes (Table 2). Tables 3 and 4 show the fertilisation scheme of the N (Table 3) and K treatments (Table 4).

**Table 3.** The fertilisation scheme of the N-treatments and the equivalent amount of nutrients applied in the different treatments of the pot experiment.

| | The Total Application Rate On each Pot | | | | | | Equivalent to Elements Applied to the Field | | | |
|---|---|---|---|---|---|---|---|---|---|---|
| | CAN | Kali60 | MgSO$_4$ | TSP | LFD | AS | N | K$_2$O | P$_2$O$_5$ | SO$_3$ |
| Treatments | g pot$^{-1}$ | g pot$^{-1}$ | g pot$^{-1}$ | g pot$^{-1}$ | g pot$^{-1}$ | g pot$^{-1}$ | kg ha$^{-1}$ | kg ha$^{-1}$ | kg ha$^{-1}$ | kg ha$^{-1}$ |
| CAN—0% of the advised amount N | 0 | 0.7 | 0.3 | 0 | 0 | 0 | 0 | 70 | 0 | 23 |
| CAN—20% of the advised amount N | 1.4 | 0.7 | 0.3 | 0 | 0 | 0 | 62 | 70 | 0 | 23 |
| CAN—40% of the advised amount N | 2.8 | 0.7 | 0.3 | 0 | 0 | 0 | 124 | 70 | 0 | 23 |
| CAN—70% of the advised amount N | 4.9 | 0.7 | 0.3 | 0 | 0 | 0 | 217 | 70 | 0 | 23 |
| CAN—100% of the advised amount N | 7.1 | 0.7 | 0.3 | 0 | 0 | 0 | 310 | 70 | 0 | 23 |
| LFD—40% of the advised amount N | 0 | 0 | 0.2 | 0 | 5.1 | 0 | 124 | 190 | 19 | 23 |
| AS—40% of the advised amount N | 0 | 0.7 | 0.04 | 0 | 0 | 3.4 | 124 | 70 | 0 | 23 |

CAN: calcium ammonium nitrate; Kali60: potassium chloride 60% K$_2$O; MgSO$_4$: magnesium sulphate; TSP: triple superphosphate; LFD: liquid fraction of digestate; and AS: ammonium sulphate solution.

**Table 4.** The fertilisation scheme of the K-treatments and the equivalent amount of nutrients applied in the different treatments of the pot experiment.

| | The Total Application Rate on Each Pot | | | | | | Equivalent to Elements Applied to the Field | | | |
|---|---|---|---|---|---|---|---|---|---|---|
| Treatments | CAN | KCL Solution | MgSO$_4$ | TSP | LFD | KC | N | K$_2$O | P$_2$O$_5$ | SO$_3$ |
| | g pot$^{-1}$ | g pot$^{-1}$ | g pot$^{-1}$ | g pot$^{-1}$ | kg ha$^{-1}$ | g pot$^{-1}$ | kg ha$^{-1}$ | kg ha$^{-1}$ | kg ha$^{-1}$ | g pot$^{-1}$ |
| KCl solution at 0% of advised amount K | 7.1 | 0 | 0.3 | 0.8 | 0 | 0 | 310 | 0 | 60 | 23 |
| KCl solution at 20% of advised amount K | 7.1 | 0.5 | 0.3 | 0.8 | 0 | 0 | 310 | 53 | 60 | 23 |
| KCl solution at 40% of advised amount K | 7.1 | 1.1 | 0.3 | 0.8 | 0 | 0 | 310 | 106 | 60 | 23 |
| KCl solution at 70% of advised amount K | 7.1 | 1.9 | 0.3 | 0.8 | 0 | 0 | 310 | 186 | 60 | 23 |
| KCl solution at 100% of advised amount K | 7.1 | 2.7 | 0.3 | 0.8 | 0 | 0 | 310 | 265 | 60 | 23 |
| LFD at 40% of advised amount K | 5.2 | 0 | 0.3 | 0.7 | 2.8 | 0 | 310 | 106 | 60 | 23 |
| KC at 40% of advised amount K | 6.9 | 0 | 0.1 | 0.5 | 0 | 3.8 | 310 | 106 | 60 | 23 |

CAN: calcium ammonium nitrate; Kali60: potassium chloride 60% K$_2$O; MgSO$_4$: magnesium sulphate; TSP: triple superphosphate; LFD: liquid fraction of digestate; and AS: ammonium sulphate solution.

The pot experiment was carried out between May and September 2020. During the growing season, field conditions were mimicked by cultivating potatoes in a semi-open netting tunnel, letting the potatoes grow in a pot filled with the soil of the potato field, and applying fertiliser at two moments in time. The potatoes were planted in 12 L PVC pots with a 28 cm diameter. Each pot had three drainage holes at the bottom. The soil moisture content was kept constant at each pot. The soil moisture content at the start of the experiment was assessed by weighing 80 g of soil and drying it in the oven under 105 °C for 24 h. The moisture content of the soil of the potato field corresponded to 22% and of the K-poor soil to 10%. Secondly, the water holding capacity (WHC) of the soil was measured

by weighing the fresh soil and then weighing it again after saturation was reached. The potato field soil was at 75% of the maximum WHC and therefore this soil was kept at 75% of the maximum WHC throughout the experiment. The K-poor soil was at 34% of the maximum WHC and therefore this soil was kept at a standard 60% of the maximum WHC. Before the pot filling, the soils were sieved through a 10 mm screen to remove the roots and debris. All pots were filled with 10 kg fresh soil and the first fertilisation dose was applied: approximately 50% of the total N amounts mentioned in Table 4 and 100% of the amount of Kali60 and $Mg_SO_4$ for the N-treatments and 50% of the amounts of K and 100% of the other fertilisers for the K-treatments. The first fertilisation dose of the BBFs was 4.8 g AS $pot^{-1}$ and 69.4 g LFD $pot^{-1}$ (equal to 59 kg N $ha^{-1}$ and 52 kg N $ha^{-1}$) for the N-treatments, and 45.3 g LF $pot^{-1}$ and 5.4 g KC $pot^{-1}$ (equal to 51 kg $K_2O$ $ha^{-1}$ and 19 kg $K_2O$ $ha^{-1}$) for the K-treatments. The soil, fertilisers, and additional water (required for the K-treatments only) were mixed thoroughly before the pot was filled. Each treatment had four replicates, which resulted in a total of 56 pots.

On 11 May 2020, potato tubers were planted in the pots. A hole was drilled in the middle of each pot (diameter of 5 cm; depth of 8 cm). One intact potato tuber (weight of approximately 65g; diameter of about 5cm) was put into the hole and the hole was then filled with soil. The overall weight of each pot was recorded. After emergence of the potatoes, the soil moisture content was checked regularly to keep the soil at 75% or 60% of its WHC. The second fertilisation took place on 19 June 2020, at the beginning of the tuber bulking stage. The second fertilisation dose of the BBFs was 5.1 g AS $pot^{-1}$ and 98.4 g LFD $pot^{-1}$, equal to 64 kg N $ha^{-1}$ and 73 kg N $ha^{-1}$. To mimic the common technique of side-dressing, the fertiliser was applied in a circular band about 10 cm away from the stems in every pot. This way of fertiliser application minimises the direct contact of plant roots and fertilisers, which can reduce the risk of salt stress. During the plant growing period, besides daily irrigation, the location of the pots was randomised every two weeks. This is because the pots at different places in the netting tunnel received different intensities of sunlight. Once a week, a chemical spraying to prevent Phytophthora infection was conducted. The potatoes were harvested at the moment that the leaves of one plant wilted completely because it indicated that all plants had an equal number of days for nutrient uptake. The plants were harvested on 10 and 11 August 2020. The fresh and underwater weight of the tubers, and the N-content in the tubers were measured; in addition, the size, the number of potatoes, and other remarks (e.g., black spots) were noted.

## 2.5. Applicability of Bio-Based Fertilisers in a Full-Scale Field Trial

Two of the BBFs (LFD and AS) were tested in a full-scale field trial in Eersel (Figure 2). The soil characteristics of the field and the fertiliser recommendations are reported in Table 2. Before the growing season, the field was ploughed using a Lemken Juwel 8 plough and scanned using a Dualem 21 (H)S soil scanner. All treatments, except the control, received 100% of the recommended amount of N (310 kg N/ha) to analyse the maximum performance of the BBFs. Additionally, all treatments, including the control, received 23 kg $SO_3$/ha, 70kg $K_2O$/ha, and 75 kg CaO/ha, which was recommended for potato growing (Table 2). The LFD and AS were tested in combinations applied with and without a base application of pig manure. This resulted in six treatments: (1) MF CAN and manure, (2) LFD and manure, (3) AS and manure, (4) only LFD, (5) only AS, and (6) a blank control with no application of N fertiliser. Each treatment had 3 replicates, which resulted in a total of 18 plots. The plots with and without manure treatment were arranged randomly and the surface area of one plot was approximately 0.75 ha (Figure 2). Manure (80 t $ha^{-1}$) was applied at the left side of the field at the beginning of the growing season using a drag hose injector (April 2020). The composition of the manure is given in Table 1. At the same moment in time, the fields with only LFD or AS received 31 t $ha^{-1}$ LFD and 2 t $ha^{-1}$ AS, which corresponds to the amount of N that the manure-treated plots received. AS was applied using a Duport Liquiliser and LFD was applied using a precision agriculture tank Tandem Premium Line in combination with the Terraject Disc.

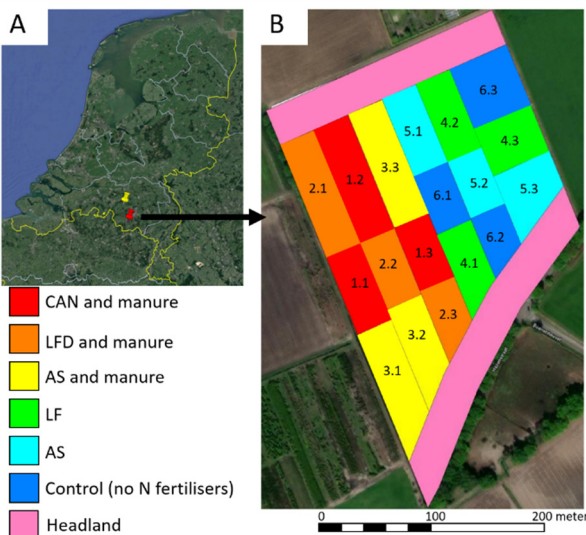

**Figure 2.** The location where the bio-based fertilisers were produced (Oirschot, yellow dot), and the location where the field experiment took place (Eersel, red dot) (**A**), together with the location of the different replenished treatments in the field (three repetitions) (**B**).

On April 29th, the consumption potatoes (variety Fontane, size 40–50mm) were planted using a Miedema CP42 planter. The distance between the planting rows was 75 cm and the distance between the potatoes in the row was about 30 cm. The second fertilisation took place on 11 June, and was applied at the beginning of the tuber bulking stage. Both BBFs were applied between the ridges using a drag hose system to ensure the tuber did not have direct contact with the fertiliser which can cause burning. Besides, fertiliser recommendations for K (70 kg ha$^{-1}$) and S (23 kg ha$^{-1}$) were met by complementing the fertilisation with Kali60 and $MgSO_4$ where needed. Manual test probing and drone flights to obtain thermal and NIR images took place at three moments during the growing season to monitor the plant growth. The test probing took place on 19 June, 28 July, and September 8, and the thermal and NIR images were taken on 21 June, 19 July, and August 7. The potatoes were harvested on October 14. After harvest, the underwater weight, number, and size of the tubers, and net yield were measured using a robot that was specifically designed for the potato farmer.

The $NO_3$-N residue in the soil profile (0–90 cm) was measured after harvest. We assume this to be a good indicator for potential N leaching to ground and surface water, which causes a serious environmental risk in the Netherlands. To measure $NO_3$-N residue in the soil profile, homogenised soil samples were taken per plot at three depths (0–30 cm, 30–60 cm, and 60–90 cm) using an auger. A 5-point sampling strategy (the centre and the 4 corners) was used to obtain a representative soil sample from each plot. The samples were collected in polyethylene sampling bags and transported from the crop field to the laboratory and stored in the freezer (−18 °C) until the analysis. The collected soil samples were extracted with 1M KCl and measured colourimetrically by a continuous flow auto-analyser (Skalar Chemlab System 4) for $NO_3$-N.

*2.6. Statistics*

The results were analysed using one-way ANOVA and Tukey's honestly significant difference (HSD). The effects of tested fertilisers were compared with the treatments and also against the used reference treatments. To investigate correlations between variables, Pearson's correlation analysis was used. Using the SPSS 22.0 software for Windows, all tests were run at a probability (*p*) level of 0.05.

## 3. Results and Discussion

### 3.1. N Release Rate of Bio-Based Fertilisers

At the start of the incubation, all tested products were applied at the same rate of 170 kg total N ha$^{-1}$ and most of the mineral N was present in the form of $NH_4^+$-N. However, due to nitrification, the concentration of $NH_4^+$-N was heavily reduced for all the treatments at day 20 and went down further below detection limits by day 40. As the concentration of $NH_4^+$-N was negligible after day 40, the total mineral N followed the same trend as the production of $NO_3^-$-N.

The average net N release ($N_{rel,net}$ calculated using Equation (1)) amounted to $140 \pm 20\%$ for AS, $113 \pm 24\%$ for LFD, $54 \pm 15\%$ for KC, and $105 \pm 15\%$ for CAN throughout the incubation (Figure 3). Net N release of AS and LFD was in general comparable with the one of CAN as a result of the high $NH_4$-N/total N ratio of these BBFs which amounted to 100% and 82%, respectively [12]. As AS solution is 100% in mineral N form, the observed net N mineralisation is seen as a result of the positive priming effect of AS on OM present in the soil [26], hence resulting in N-release value >100%. A similar result was observed in the LFD treatment from day 80 until the end of the experiment when N release values above 100% were measured. The average net N mineralisation ($N_{min,net}$, calculated using Equation (2)) from amended treatments throughout incubation duration was $40 \pm 14\%$ for AS, $36 \pm 12\%$ for LFD, and $20 \pm 3\%$ for KC (Figure 3). In general, the values were statistically different between the products ($p < 0.05$). For instance, on day 100, the values of KC and CAN are significantly different from AS and LFD (Figure 3).

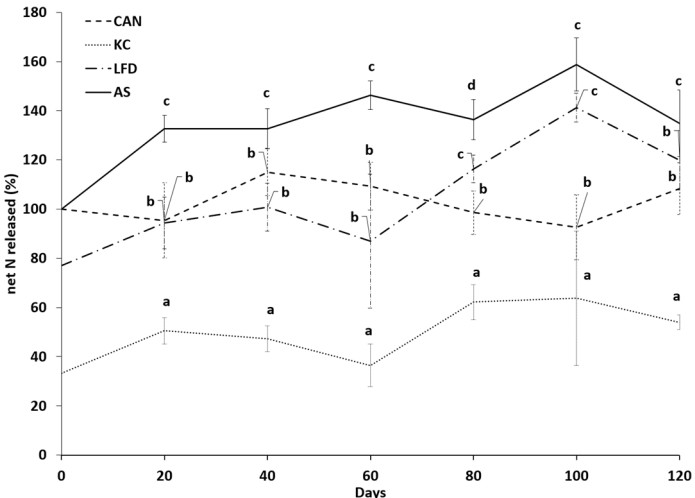

**Figure 3.** Net N release (in % of total N applied) for applied fertilisers over a time span of 120 days. Legend: AS—ammonium sulphate, LFD—liquid fraction of digestate, KC—potassium concentrate, and CAN—calcium ammonium nitrate. Lower case letters (a–d) denote statistically significant differences in means (Tukey's Test for $p < 0.05$) among the products for each sample time (t = 20, 40, 60, 80, 100, and 120).

In the case of KC, a positive net mineralisation is observed due to mineralisation of organic N present in the product. However, the net N release was inhibited throughout the incubation period. This is due to the C/N ratio > 7 and $NH_4^+$-N/total N < 50%. This relatively high C/N ratio and lower $NH_4^+$-N/total N promotes a very slow release of total N applied, reaching a stagnant release after some days [17,27]. This hypothesis could be cross confirmed from Appendix B, where Pearson correlation analysis shows a strong inverse interaction between % N released and $C/N_{total}$ ratio (r = −0.999, $p$ = 0.03). Additionally, it is also mentioned in previous studies that the C/N and $NH_4^+$-N/$N_{total}$ ratios of applied products typically influence the $N_{rel,net}$ and $N_{min,net}$ (%) [20,28–30]. However, in this case, even if the Pearson correlation effect of $NH_4^+$-N/$N_{total}$ and $N_{rel,net}$ (or $N_{min,net}$) (%) is strong (r = 0.999, $p$ = 0.023), suggesting a direct relation between variables, note that there

can be some bias caused by the relatively high initial $NH_4$-N concentrations of AS and LFD compared with the $NH_4$-N concentrations of K [31].

### 3.2. Emissions from the Bio-Based Fertilisers

The $N_2O$ emissions of all three BBFs were measured over 18 days and compared with mineral fertilisers (i.e., CAN and urea). In the mesocosm, the principle of production of $N_2O$ follows simultaneous nitrification (oxidation of $NH_4^+$ to $NO_3^-$ via $NO_2^-$) and denitrification (reduction of $NO_3^-$ to $N_2O$ and $N_2$) [32]. Following this process, the mineral fertilisers showed highest emission of $N_2O$ (0.11 ± 0.02% for urea and 0.11 ± 0.01% for CAN) (Figure 4A). This is because of the rapid hydrolysis of products after application, resulting in increased $NH_4$ availability. This further led to nitrification followed by denitrification, resulting in $N_2O$ production. This could also be observed in the case of AS (0.03 ± 0.008% of AS N applied) and LFD (0.02 ± 0.005% of LFD N applied), where the initial concentration of mineral N was ~100% and ~80% of total N (Table 5). Additionally, a similar result to CAN is observed for KC, where emission of 0.05 ± 0.03% of KC N applied could be observed. These results for KC are expected due to the combination of two factors—a moderate NH4 percentage at the initial application (~33%), and a high TOC content in the fertiliser (Table 1). As the OC serves as an energy source for denitrification, the overall oxygen content in the soil decreases, promoting denitrification of $NH_4$ in BBFs [33]. This shows a strong correlation between the $N_2O$ emissions and initial ammonium ($NH_4^+$) content which is also supported by different studies [20,28–30]. Therefore, highest $N_2O$ emissions were observed in the soils that were treated with mineral fertiliser, which is also confirmed by [34].

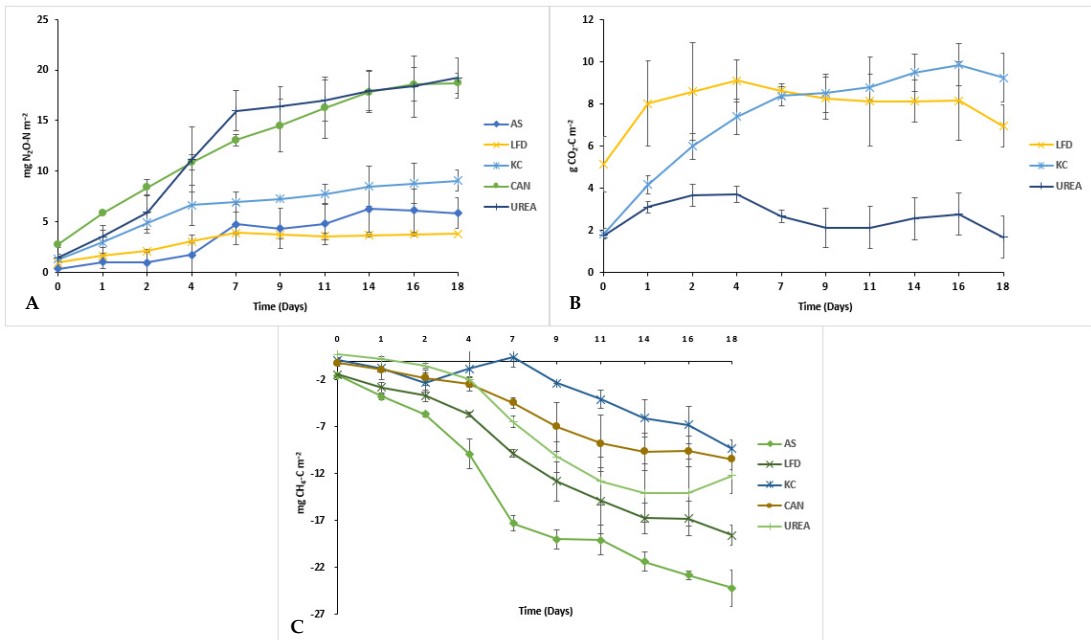

**Figure 4.** Cumulative emissions of nitrous oxide (mg $N_2O$-N per $m^2$ of soil) (**A**), carbon dioxide (g $CO_2$-C per $m^2$ of soil) (**B**), and methane (mg $CH_4$-C per $m^2$ of soil) (**C**) emissions measured during incubation of digestate-derived organic fertilisers. Legend: AS—ammonium sulphate, LFD—liquid fraction of digestate, KC—potassium concentrate, and CAN—calcium ammonium nitrate.

The $CO_2$ flux developed in the soil is mainly produced by the respiration of soil microorganisms and dead plant roots. The $CO_2$ emissions from the mesocosms are directly dependent on the initial OC content available in the BBFs and mineral fertilisers. The KC and LFD had the highest amount of OC resulting respectively in emissions of 8.97 ± 3% and 25.9 ± 5% of total C added from the product. Urea also consisted of 20% carbon and showed rapid mineralisation due to hydrolysis, hence resulting in around 60% of $CO_2$

emissions within the first five days of the experiment. In general, organic fertilisers produce considerably higher outcomes because they enhance the fraction of OC in the soil, which is more readily available for microorganisms in respiration [35]. Therefore, the released C from BBFs in soils is seen as biogenic C and is considered to be C-neutral as it does not contribute to the net $CO_2$ increase. The results for AS and CAN are not shown in Figure 4B as a negligible amount of OC is present in both cases to cause any $CO_2$ emissions. Any type of $CO_2$ emissions seen in these products is due to the positive priming effect of the already available carbon present in the soil.

**Table 5.** Emission factors of nitrous oxide ($N_2O$), carbon dioxide ($CO_2$), and methane ($CH_4$) expressed per 100 g of N applied for tested bio-based fertilisers and mineral N fertilisers.

| Fertiliser Type | $N_2O$ | $CO_2$ | $CH_4$ * |
|---|---|---|---|
| | per 100g N Applied | | |
| Liquid fraction of digestate | $0.02 \pm 0.005$ | $53 \pm 9$ | $-0.11$ |
| Potassium concentrate from evaporator | $0.05 \pm 0.03$ | $58 \pm 9$ | $-0.06$ |
| Ammonium sulphate solution | $0.03 \pm 0.008$ | $-20 \pm 12$ | $-0.14$ |
| Calcium ammonium nitrate | $0.11 \pm 0.01$ | $-20 \pm 7$ | $-0.06$ |
| Urea | $0.11 \pm 0.02$ | $20 \pm 6$ | $-0.08$ |

* The standard deviations were not significant considering the relatively lower values of $CH_4$.

Across the study, the $CH_4$ emissions were significantly low due to the presence of aerobic conditions during the course of incubations (Figure 4C). Moreover, the application of manure-derived products on soil proved to enhance soil aeration, hence reducing $CH_4$ emissions [35]. The phenomenon of methanogenesis and methanotrophy derives the net soil $CH_4$ flux [36]. In the case of all BBFs as well as mineral fertilisers (CAN and urea) the $CH_4$ uptake is higher than $CH_4$ production, hence resulting in negative methane emissions from the soil.

The GHG emission factors (EFs) were computed for $CH_4$, $CO_2$, and $N_2O$ (Table 5). The EFs of tested fertilisers for $N_2O$ were in the range of 0.02–0.11 g per 100 $g^{-1}$ N applied which is lower than the IPCC Guidelines for National Greenhouse Gas Inventories [21]. The IPCC default value of direct $N_2O$ emissions is equivalent to 1% of total N applied on the soil. However, the IPCC guidelines state EFs from the data culminated over the year, and this study focuses on the short-term emission of 18 days which is one of the reasons for lower EFs. In a study on short-term GHGs [37], EFs of 0.1–0.49% for urea applied at 200 kg N $ha^{-1}$ were found, which were quite similar to the results in this study. A review study [38] indicated that the $N_2O$ emission factors from urea range between <0.1 to about 2% of applied N.

From Table 5, AS has shown the least emissions across all three gases. Moreover, LFD has performed better than MFs in the case of $N_2O$ and $CH_4$. From these results, we can conclude that the GHG emissions of BBFs resulting from the secondary treatment of raw manure (AS) are less than that of untreated/primary treated animal manure (i.e., LFD and KC).

### 3.3. Nutrient Fertiliser Replacement Value

A response yield curve was fitted through the measured crop yields that correspond to different mineral N or K fertiliser application rates. The curve of the mineral N fertiliser fit more nicely in a polynomial trendline ($R^2 = 0.99$) compared with the yield response curve of the mineral K fertiliser ($R^2 = 0.92$) (Figure 5). The fresh yield at 40% of the recommended N rate was 0.43 kg $pot^{-1}$ for the potato plants that received mineral N fertiliser, and slightly higher (0.47 and 0.49 kg $pot^{-1}$) for the potato plants that received LFD and AS, respectively. The yield at 40% of the recommended K rate was 0.33 kg $pot^{-1}$ for the plants that received mineral K fertiliser, and 0.42 and 0.41 kg $pot^{-1}$ for LFD and KC, respectively. The corresponding NFRV of N treatments was 1.04 for LFD and 1.13 for AS. For the K treatments, the NFRV was 1.52 for LFD and 1.41 for KC.

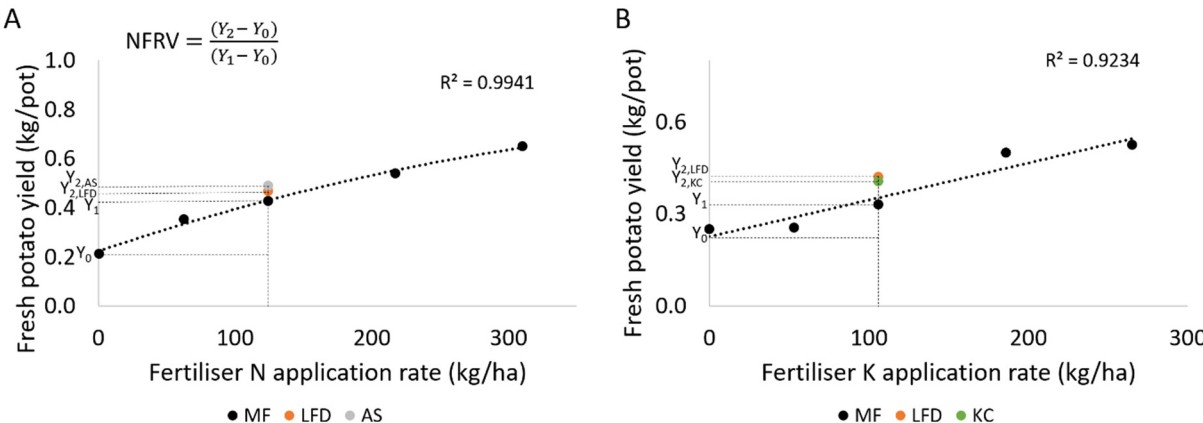

**Figure 5.** The yield response curves of mineral fertiliser N (**A**) and K (**B**) together with the performance of the liquid fraction of digestate (LFD), the ammonium sulphate solution (AS), and the K-concentrate (KC). $Y_0$ is the yield obtained from the yield curve when no mineral fertiliser is applied, $Y_1$ is the yield obtained from the yield curve when a fixed amount of mineral fertiliser is applied, and $Y_2$ is the yield obtained from the treatment that received BBF.

The potato yield of the plants that received BBFs are in all cases slightly higher compared with the mineral fertiliser, in which it turned out that all NFRVs are above the response yield curve. Other studies showed that often only assessments on long-term NFRV reach yields above the response yield curve [22]. However, because the total potato yield per pot is low, slightly higher yields (in our case between 40 g and 90 g) can result in NFRV > 1. The values should therefore be evaluated with caution. This experiment showed that (i) both BBF perform well in comparison with mineral fertiliser, and (ii) the level of refinement does not seem to influence the NFRV.

Other characteristics of the potato also do not show significant differences between the treatments with a highly refined BBF, a less refined BBF, or mineral N or K fertiliser. The size and number of tubers did not differ much between the treatments, but within a treatment sometimes a large variation exists. The '40% K concentration using LFD'-treatment counted, for example, between 1 and 12 potatoes smaller than 3 cm. The total N concentration in the tubers differed significantly, but the total K concentration in the tubers did not differ significantly. The N-content in the potato tubers increased from $6.5 \pm 0.3$ g kg$^{-1}$ DM when 0 kg ha$^{-1}$ N was applied to $11.3 \pm 0.4$ g kg$^{-1}$ DM when 310 kg N ha$^{-1}$ was applied. A N application rate of 40% of MF resulted in a total N content in the potato tuber of $8.9 \pm 0.7$ g kg$^{-1}$ DM. A N application rate of 40% in the form of LFD and AS resulted in a total N content in the potato tuber of $8.1 \pm 1.0$ g kg$^{-1}$ DM and $8.7 \pm 1.0$ g kg$^{-1}$ DM, respectively. Calculating the NFRV based on the N uptake in the potato tuber resulted in a NFRV of 0.50 for LFD and 0.60 for AS. Similar to the NFRV resulting from the fresh potato yield, the NFRV resulting from the N uptake in the tuber was slightly higher in the more refined AS product compared with the less refined LFD product. The slower release of N by LFD compared with AS or mineral N fertiliser can clarify this lower NFRV.

### 3.4. Applicability of Bio-Based Fertilisers in a Full-Scale Field Trial

At the start of the growing season, a soil scan was carried out to provide insight in the spatial variation in the soil electric conductivity (EC) within the experimental field. The average EC per plot at 1 m depth ranged between 10.0 and 13.2 mS m$^{-1}$, with highest values in the north and west and lowest values in the south and east (Figure 6B). A gas pipe line was identified as a straight line in the upper part of the field. At three moments during the growing season thermal and NIR images of the field were collected. The weighted (near-infrared-red) difference vegetation index (WDVI) image taken at the beginning of the growing season showed little difference in the treatments that received manure (left side of the field), and the control plots are clearly visible (Figure 6C). The other NIR images showed

similar results. The thermal map taken at the end of the growing season showed clear indications of heat stress (Figure 6D). The plots treated with AS had a lower temperature compared with the other plots. Again, no clear pattern was visible in the plots that received manure. The control plots showed clearly lower yields ($60 \pm 13.6$ t ha$^{-1}$) compared with the other plots (Figure 6E). Highest yields were obtained in the plots that were treated with AS ($73 \pm 9.1$ t ha$^{-1}$) followed by the plots treated with manure and mineral fertiliser ($68 \pm 5.9$ t/ha). The other treatments showed quite similar results ($65 \pm 7.8$ t ha$^{-1}$ for LFD, $65 \pm 4.1$ t ha$^{-1}$ for the manure and AS treatment, and $63 \pm 1.7$ t ha$^{-1}$ for the manure and LFD treatment).

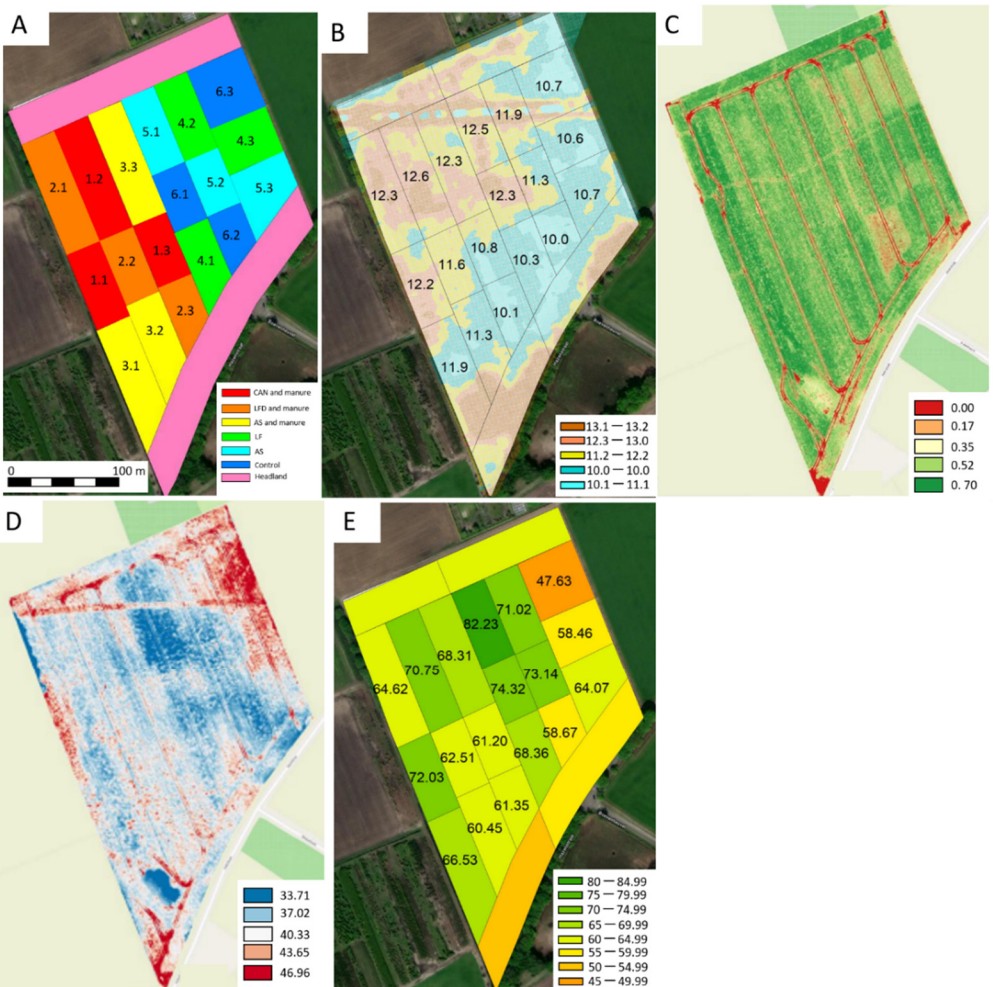

**Figure 6.** Summary of the field performance during the growing season. The different treatments and repetitions (**A**), the electric conductivity (mS m$^{-1}$) (**B**), the weighted (near-infrared-red) difference vegetation index (-) measured on 21 June 2020 (**C**), the thermal heat map (°C) measured on 7 August 2020 (**D**), and the average potato yield per plot (t ha$^{-1}$) (**E**).

For the market, consumption potatoes need to have an underwater weight between 70 g and 80 g kg$^{-1}$. The treatments with only LFD are at the lower part of this range (72.2 g kg$^{-1}$), the treatments with mineral N fertiliser are at the upper part of this range (80 g kg$^{-1}$), and the control treatment (no mineral N fertiliser) is far above (92 g kg$^{-1}$). Looking at the size distribution of the potato tubers, the control treatment that received no N fertiliser and the treatment with manure and LFD had a large number of small potato tubers compared with the other treatments. In these treatments, 70%, in the case where no N fertiliser applied, and 59%, in the case where manure and LFD was applied, had a size smaller than 5 cm of the total numbers of potato tubers counted during test probing. In the mineral N fertiliser-treated plots only 39% of the counted potato tubers had a size smaller

than 5 cm during the test probing. For the other treatments, the size of the potatoes were comparable with the treatment with mineral N fertiliser. The number of tubers that were counted during test probing were lowest in the plot that received LDF only (45 tubers). The number of tubers for the other treatments were very similar and ranged between 57 for the treatment with LFD and manure, and 66 for the treatments with AS and manure. These results illustrate the slightly poorer performance of LFD compared with AS.

After harvest, the $NH_4$-N and $NO_3$-N residues in the soil were analysed (Figure 7). The potential for residual nitrate leaching to the ground and surface water is one of the important aspects of the safe application of BBFs. The residual nitrate in the soil profile (0–90cm) in the post-harvest period showed no significant difference in leaching risk for BBFs in comparison with the used mineral fertiliser. However, all the treatments (including the unfertilised control) resulted in relatively high nitrate residues. In the Netherlands, there is no legal limit on maximum allowable nitrate residue. On the other hand, in Flanders, in the neighbouring region, nitrate residue in potato cultivation should not exceed (depending on the location of the field) the maximum limit of 165 kg $NO_3$-N ha$^{-1}$ according to current Flemish environmental standards (VLM, 2021). In general, high nitrate residue is considered common in the case of potatoes due to their ability to uptake only 50–60% of the applied N [39], and also their shorter root depth [40]. Additionally, the effect of rainfall during the growing season is also found to be inversely proportional to the nitrate residue in 0–90cm soil depth [39]. The high application of N (310 kg ha$^{-1}$) resulted in high residue concentrations in the soil. The $NO_3$-N residue in the soil profile was on average highest in the plots that only received AS (264 kg ha$^{-1}$), but the residues in plots that received mineral fertiliser and manure (256 kg ha$^{-1}$), LFD and manure (258 kg ha$^{-1}$), and AS and manure (245 kg ha$^{-1}$) were almost as high. The $NO_3$-N residue in the soil profile was, except from the control, lowest in the plots that only received LFD.

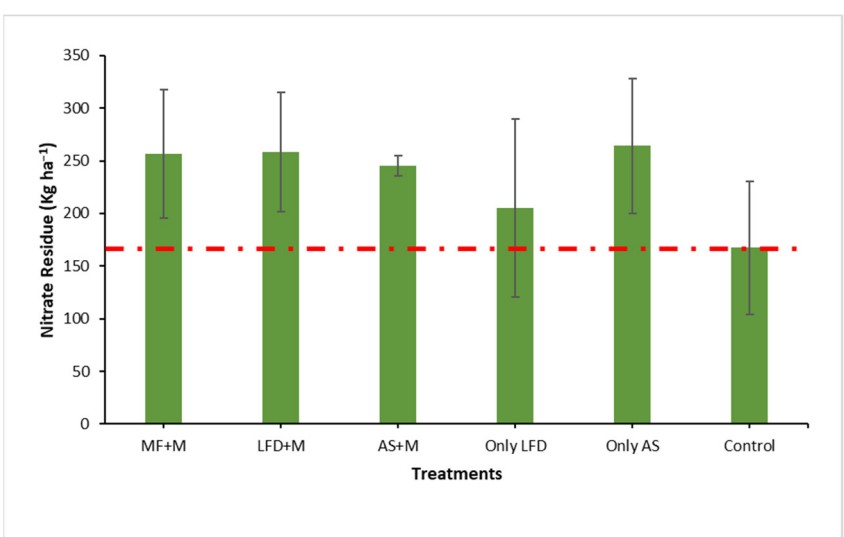

**Figure 7.** Amount of nitrate residue (kg ha$^{-1}$ in 0–90 cm soil depth) in post-harvest period for five treatments (MF + M = CAN + manure, LFD + M = liquid fraction of digestate and manure, AS + M = ammonium sulphate solution and manure, only LFD = standalone treatment from liquid fraction of digestate, and only AS = standalone treatment from ammonium sulphate solution). Red dotted line represents the legislative limit (165 kg ha$^{-1}$) according to Flemish environmental standards.

*3.5. Environmental and Agronomic Evaluation*

This study showed that all three BBFs are safe to use in potato growing from an agronomic and environmental perspective (Figure 8). Possibly, the effect can be influenced more strongly by soil type, e.g., soil pH, rather than by quantity and quality of the applied BBF [41]. Summarized, after soil application, AS has a relatively low $N_2O$ emission factor. It has highest N release rate and NFRV, and also in the field trial the N concentration in

the potato tuber, the potato yield, and the $NO_3$-N residue in the soil after harvest were comparable with the results of the MF-treated plots. The potato yield of the plot that only received AS was highest of all plots. However, in the Netherlands it is not very likely that AS is applied without manure. AS is tested for being classified as RENURE product [13]. Acidification of the soil is a concern of using AS frequently. This can have negative impacts on mineral composition and biodiversity [34].

| | AS | LFD | KC | MF |
|---|---|---|---|---|
| **EF for GHGs (per 100g N applied)** | $N_2O$: 0.03±0.008 | $N_2O$: 0.02±0.005 | $N_2O$: 0.05±0.03 | $N_2O_{UREA}$: 0.11±0.02<br>$N_2O_{CAN}$: 0.11±0.01 |
| **N dynamics from incubations** | N release rate: 142±19% | N release rate: 113±24% | N release rate: 53±16% | N release rate: 105±16% |
| **NFRV** | $NFRV_N$: 1.13 | $NFRV_N$: 1.04<br>$NFRV_K$: 1.52 | $NFRV_K$: 1.41 | $NFRV_{MF}$: 1.00 |
| **Field trial** | N in tuber: 8.7±0.97 g $kg^{-1}$ DM<br><br>$Yield_{AS}$: 73±9.1 t $ha^{-1}$<br>$Yield_{AS+man}$: 65±4.1 t $ha^{-1}$<br><br>$NO_3$-N $residue_{AS}$: 264±64 kg $ha^{-1}$<br>$NO_3$-N $residue_{AS+man}$: 245±10 kg $ha^{-1}$ | N in tuber: 8.1±0.97 g $kg^{-1}$ DM<br><br>$Yield_{LFD}$: 65±6.6 t $ha^{-1}$<br>$Yield_{LFD+man}$: 63±1.7 t $ha^{-1}$<br><br>$NO_3$-N $residue_{LFD}$: 205±85 kg $ha^{-1}$<br>$NO_3$-N $residue_{LFD+man}$: 258±56 kg $ha^{-1}$ | Not tested in field trial | N in tuber: 8.9±0.68 t $ha^{-1}$<br><br>$Yield_{MF+man}$: 68±5.9 t $ha^{-1}$<br><br>$NO_3$-N $residue_{MF+man}$: 256±61 kg $ha^{-1}$ |

**Figure 8.** Summary of the results to compare refined bio-based fertilisers (ammonium sulphate (AS) and potassium concentrate solution (KC)) with less refined bio-based fertiliser (liquid fraction of digestate (LFD)) and mineral fertiliser (MF) with and without manure (man) on: (i) greenhouse gas (GHG) emission factor (EF) (blue), (ii) N release rate (yellow), (iii) nutrient fertiliser replacement value (NFRV), and (iv) potential field application (green).

The $N_2O$ emission after the application of LFD was low. The N release rate and the NFRV are lower compared with the highly refined BBF AS, but higher compared with MF. In the field trial, the plots that were treated with LFD showed lowest N concentration in the tuber and lowest potato yield, and performed poorest in the qualitative assessment of the potatoes after harvest. In practice, farmers are restricted to the amount of N from livestock manure they can apply on the field (170 kg N/ha). A meta-data analysis showed that LFD met the criteria of RENURE in only 43% to 58% of the analysed data [42]. Therefore, it is not very likely that LFD will become a RENURE product. However, it can serve as a replacement for slurry manure in areas where the application is restricted by the P regulations because the P content in LFD is lower (0.7 instead of 2.4 g $kg^{-1}$). However, the adoption of LFD will depend on the price farmers receive or pay for the product. Long-term application of LFD can build up nutrients in the soil which can improve the NFRV [22]. It is recommended to investigate the effect of BBF refinement on long-term application or on the use of precision fertigation.

$NH_3$ emissions were not measured, but it is expected that $NH_3$ emissions are low during the application of AS because of the low pH of the product. The farmer had problems applying the small amounts of AS to the plots and prefers to dilute the product. $NH_3$ emissions increase when the pH of the product increases above pH 5 and should therefore be taken into account when AS is diluted. The $NH_3$ emissions during the application of LFD depend strongly on the application method [12]. Confirmed by a fertiliser machinery developer in the Netherlands, machinery for injecting LFD during the potato growing season does not exist. The dispensing nozzles will be clogged by the fibres in the LFD. Therefore, surface application of LFD is unavoidable at the moment.

The highly refined BBF KC showed moderate $N_2O$ emissions. The N release rate is, as expected, the lowest of all fertiliser products tested. The NFRV is higher compared with MF, but lower compared with the less refined BBF LFD. Because of the high K-concentration in

the soil, this product was not tested in the field trial. The use of K-concentrate is expected not to change total $NH_3$ and $N_2O$ emissions in the Netherlands, but during the production process $NH_3$ emissions take place [43].

Refined products are more costly to produce compared with less refined products. In this study, the production costs of LFD is approximately 3-euro tonne$^{-1}$ digestate and the production of AS is approximately 5.4-euro tonne$^{-1}$ digestate in this study. However, because LFD and AS are seen as waste products, they were sold for −15- and −5-euro tonne$^{-1}$, respectively, in this trial. This will change when LFD and/or AS are listed as the RENURE product or when the manure surplus in the Netherlands decreases. For MF, the potato farmer is paid 180 euro tonne$^{-1}$ and for manure the farmer receives 15 euro tonne$^{-1}$. The N-concentration of LFD is much lower (3.7 g kg$^{-1}$) compared with AS (81.6 g kg$^{-1}$) or MF (260 g kg$^{-1}$), and therefore larger amounts of LFD are needed to apply equal amounts of N (Table 6). The application of LFD was labour intensive and machinery needed to be adapted because high amounts of LFD were applied on small plots. These results are case specific and should therefore not be applied to other cases. Although some practical issues need to be solved, the use of BBFs in arable farming is profitable at the moment. This was also confirmed by other studies [20]. However, policy restrictions on the use of LFD and AS hamper the adoption of BBF as a replacement for mineral fertiliser at the moment.

**Table 6.** The purchase and application costs of the bio-based fertilisers liquid fraction of the digestate (LFD) and Ammonium Sulphate (AS) and the mineral fertiliser CAN.

| | Quantity Required | Product Costs | Application Costs | Total Costs |
|---|---|---|---|---|
| | (kg/ha) | (€ kg$^{-1}$ N Application ha$^{-1}$) | (€ kg$^{-1}$ N Application ha$^{-1}$) | (€ kg$^{-1}$ N Application ha$^{-1}$) |
| Manure + CAN | Manure: 80,000 MF: 200 | 5.7 | 30 | 35.7 |
| Manure + LFD | Manure: 80,000 LFD: 10,900 | −4.4 | 87.7 * | 83.3 * |
| Manure + AS | Manure: 80,000 AS: 600 | −3.9 | 4.8 | 0.9 |
| LFD | LFD: 64,778 | −2.5 | 321.8 * | 319.3 * |
| AS | AS: 3,582 | −1.5 | 25.7 | 24.2 |

* Trial specific due to the labour-intensive application of large amounts of LFD on small plots.

## 4. Conclusions

This study showed that all three BBFs are safe to use as a replacement for mineral fertiliser or slurry manure in potato growing on sandy soil, although there are still some practical issues related to the application of BBF to be solved. Environmentally, refined BBFs AS and KC performed, in terms of $N_2O$ emission, slightly worse compared with the less refined BBF LFD, whereas agronomically the crop yield was slightly better in the case of AS. However, in combination with manure, the BBFs AS and LFD did not show significant differences in crop yield. The hypothesis that refined BBFs perform environmentally and agronomically better compared with less refined BBFs was therefore rejected. Compared with MF, all BBF had lower $N_2O$ emissions but also slightly lower crop yields (except the field that was treated with AS only). Overall, we conclude that agricultural circularity can be stimulated by (i) solving the practical issues that occurred during the application of LFD, (ii) making sure BBFs are on the list of RENURE materials so they can legally replace mineral fertiliser, and (iii) reducing the surplus of slurry manure to stimulate the use and fair pricing of BBF products.

**Author Contributions:** Conceptualization, C.M.J.H., I.S., J.P.L., E.M. and R.v.N.; data curation, C.M.J.H., V.S., I.S. and Z.Y.; formal analysis, C.M.J.H., V.S., I.S. and Z.Y.; investigation, V.S.; methodology, C.M.J.H., V.S., I.S. and R.P.J.J.R.; visualization, C.M.J.H. and V.S.; writing—original draft, C.M.J.H. and V.S.; writing—review and editing, C.M.J.H., V.S., I.S., J.P.L., E.M., R.v.N., Z.Y. and R.P.J.J.R. All authors have read and agreed to the published version of the manuscript.

**Funding:** This research was funded by the Nutri2Cycle project who receives funding from the European Union's Horizon 2020 Framework Programme for Research and Innovation under Grant Agreement no 773682 and from the Dutch Ministry of Agriculture (AF-EU-18030).

**Institutional Review Board Statement:** Not applicable.

**Informed Consent Statement:** Not applicable.

**Data Availability Statement:** Data are available on request due to restrictions, e.g., privacy or ethical. The data presented in this study are available on request from the corresponding author. The data are not publicly available due to privacy.

**Acknowledgments:** This research was carried out in collaboration with Van den Borne Potatoes, Practice Center for Precision Agriculture, a pig farmer, and VP-Hobe. We are very grateful for their contributions.

**Conflicts of Interest:** The authors declare no conflict of interest.

## Appendix A

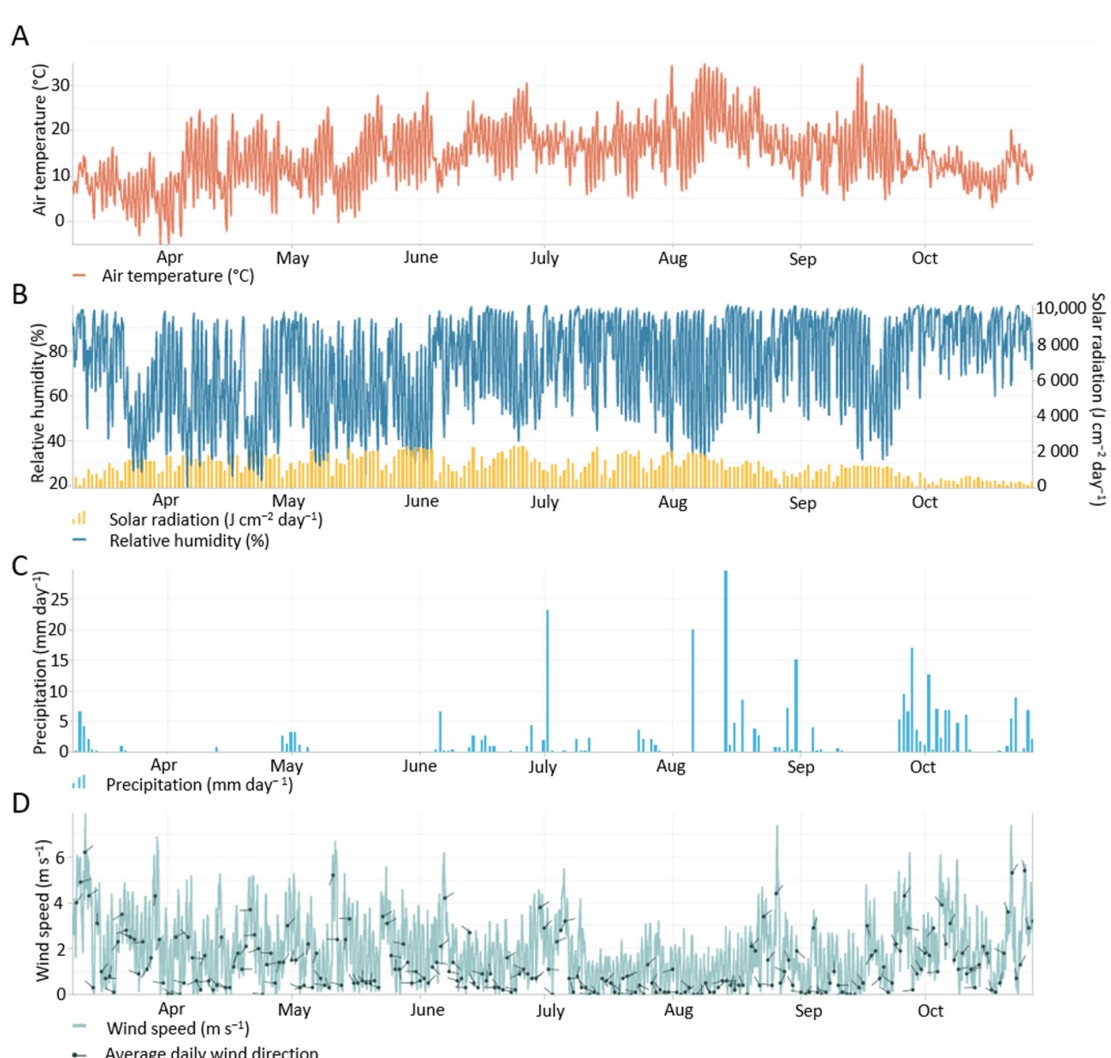

**Figure A1.** Weather data during the growing season, including the air temperature (°C) (**A**), solar radiation (J/cm$^2$) and humidity (%) (**B**), precipitation (in mm) (**C**), and wind speed (m/s) and direction (per 24 h) (**D**).

## Appendix B

**Table A1.** Pearson correlation analysis between different parameters in relation to the BBFs, N release, and N mineralization dynamics (*n* = 3).

| Parameters | OC (g kg⁻¹) | Ntot (g kg⁻¹) | NH4-N (g kg⁻¹) | C/Ntot | C/Organic N | NH4-N/Ntot | Organic N/Ntot | Total N-min (mg kg⁻¹) | N Release (%) | N Mineralization (%) | Ntot Added (mg) | NH4-N Added (mg) | OC Added (mg) |
|---|---|---|---|---|---|---|---|---|---|---|---|---|---|
| OC (g kg⁻¹) | - | | | | | | | | | | | | |
| Ntot (g kg⁻¹) | −0.558 | - | | | | | | | | | | | |
| NH4-N (g kg⁻¹) | −0.588 | 0.999 * | - | | | | | | | | | | |
| C/Ntot | 0.977 | −0.721 | −0.746 | - | | | | | | | | | |
| C/organic N | 0.711 | −0.98 | −0.987 | 0.844 | - | | | | | | | | |
| NH4-N/Ntot | −0.996 | 0.633 | 0.661 | −0.993 | −0.774 | - | | | | | | | |
| OrganicN/Ntot | 0.996 | −0.633 | −0.661 | 0.993 | 0.774 | −1.000 ** | - | | | | | | |
| Total N-min (mg kg⁻¹) | −0.962 | 0.762 | 0.786 | −0.998 * | −0.875 | 0.984 | −0.984 | - | | | | | |
| N release (%) | −0.986 | 0.688 | 0.714 | −0.999 * | −0.818 | 0.997 * | −0.997 * | 0.994 | - | | | | |
| N mineralization (%) | −0.96 | 0.769 | 0.792 | −0.997 * | −0.88 | 0.982 | −0.982 | 1.000 ** | 0.993 | - | | | |
| Ntot added (mg) | 0.528 | −0.999 * | −0.997 * | 0.696 | 0.972 | −0.605 | 0.605 | −0.739 | −0.661 | −0.745 | - | | |
| NH4-N added (mg) | −0.996 | 0.629 | 0.657 | −0.992 | −0.77 | 1.000 ** | −1.000 ** | 0.983 | 0.997 * | 0.981 | −0.6 | - | |
| OC added (mg) | 0.935 | −0.227 | −0.263 | 0.839 | 0.415 | −0.898 | 0.898 | −0.803 | −0.863 | −0.797 | 0.192 | −0.9 | - |

* Correlation is significant at the 0.05 level (2-tailed). ** Correlation is significant at the 0.01 level (2-tailed).

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
