# Peer review of "Replacing Mineral Fertilisers for Bio-Based Fertilisers in Potato Growing on Sandy Soil: A Case Study"

_applsci, doi:10.3390/app12010341_

Round 1

Reviewer 1 Report

Dear Corresponding Author,

The current manuscript is an interesting study on the use of bio-based fertilizers (BBFs) instead of chemical ones to increase the sustainability of the farming sector, in line with the circular economy and closure of the cycles approaches. In particular, the work analysed the performance of three BBFs in comparison with mineral fertilizers looking at 1) N released in an incubation experiment; 2) N2O, CO2 and CH4 emissions due to the applications of each nutrient substance in soil microcosms; 3) calculation of the so-called nutrient fertilizer replacement value (NFRV) in pot; 4) agronomical characterization in a potatoes field trial.

In my opinion, the manuscript is very interesting. In general, I found the reading of this work very nice and smooth. In my opinion, this research article is endowed of a very interesting and well-told content, presented in an accurate way, as supposed to be for being published in your journal. The reported results are rightly supported from the data collected and analyzed. Finally, the manuscript deserves to be published, it just needs little improvements. Some suggestions are listed below.

My final recommendation is acceptable after “Minor Revision”.

Introduction

It is very concise, but self-sufficient. Notwithstanding, I would suggest enriching it with information on BBFs, including some additional references besides the only one mentioned [4].

As example, you could consider at least these references (but also others, these are not self-citations):

https://doi.org/10.1186/1475-2859-13-66

https://doi.org/10.1016/j.biortech.2019.122223

https://doi.org/10.1038/s41598-019-56954-2

The refinement processes could be also briefly described.

Line 64: Delete line 64, since the same is said also in line 70.

M&M

Line 79: add a space between 1000 and m3

Line 88: you have mentioned organic matter (OM) immediately after dry matter (DM), but then in Table 1 you have put the OM related column at the end. It is suggested to edit the chemical properties appearing in Table 1 according to the order followed in the text. Since EC is a physical property, you should mention it at the end.

Line 92: insert a comma between “1 M KCl” and “conform ISO”. Moreover, please consider - throughout the text - that the numerical value always precedes the unit, and a space is always used to separate the unit from the number.

Why did you not show the values for NO3-N in Table 1? Since you have mentioned it in the text, a reader expects to see it in the Table. If you don’t show it, please add the sentence like “(data not reported in Table 1)” or something similar.

Line 93-94: (TOC) was measured. Use always the same tense (simple past in this case).

Lines 94-95: P-PRO-32 spectrophotometer… Please give further information, like the brand/company producer…?

Table 1: the reported values come from a single measure, isn’t it? Otherwise, you would have reported a standard deviation. Please, explicit also this information on the text. The same comment is addressed to Table 2, 3 and 4.

In the text you should also report how did you measure TOTAL CARBON (TC).

Line 106-107: If you put “N/A”, it is not necessary to write in the Table Legend where it is N/A, it is already visible in the table. So, in the legend, just leave “N/A: not applicable”

Line 116: delete the abbreviation ASL above sea level since it is just named once in the text.

Please translate into English all terms in Appendix A

Line 121: replace “was” with “were”

Line 134: put a space between 271 and g

Table 2: microbial biomass and activity are shown without mentioning the analytical methods. Then, you never mentioned them in the text. You give several information that is not useful and indeed it is not discussed. This kind of information maybe could be moved in the Supplementary Materials in order to keep the reading of the manuscript easier and not to distract the reader with extra-information which seem not to be related to the focus of the paper…

Results

Legend to Figure 7. Please list all the treatments/soil samples analysed. You mentioned five treatments, but you did explicit only three of them.

Line 316: “In general, the values were not statistically different between the products (p<0.05).” If p<0.05 should mean that there is a significant difference! Maybe in the abovementioned sentence you should put (p>0.05).

Lines 316-317: I cannot understand. Looking at Fig. 3, I see the following: KC remains almost constant (p>0.05); LFD shows a smooth variation at days 80 and 100 (p>0.05); AS shows a very smooth variation at day 80 (p>0.05, but looking at the graph and the error bars, it does not seem so…); CAN remain almost constant (p>0.05).

Other small inaccuracies are present. I have found throughout the text that you give several information/details that are not useful to make the story more clear or more interesting; on the contrary, they distract the reader from the important results…

Finally, I would suggest you performing just MINOR REVISIONS for publishing your manuscript.

Best regards.

Reviewer 2 Report

The manuscript (applsci-1515168) entitled “Replacing mineral fertilizers for bio-based fertilizers in potato growing on sandy soil: a case study" submitted for its publication on Applied Sciences addresses an interesting topic. The authors compared the differences between bio-based and mineral fertilizers for N release rate, greenhouse gas emission, nutrient fertilizer replacement value, potato quality and quantity and the remaining N residues in the soil after harvest. The findings can provide support for the use of BBF, AS, KC and LFD. The article

Introduction

  1. The introduction is weak, 6 referenced articles are not enough to support the background and the significance of the study. There are a lot of studies regarding to bio-based fertilizers. How is the going of the researches, what findings are clear and what should be in-depth research, what is the innovation of your study?
  2. Check the names of articles in line 55, “Recovered” “manURE”
  3. Line 66-73 are experimental design which should be put into materials and methods
  4. The objectives and hypothesis should be introduced in introduction.

Materials and methods

  1. There is too much redundant information in this section, such as line 88-101, line 241-254, please simplify it.
  2. What were tested in sentence “The BBFs were tested in a crop field of the potato farmer that is specialised in using precision agriculture”?
  3. What is the meaning for “The field has a 1:4 crop rotation”?
  4. Check the note in Line 146 (mineralisation rate (Nmin,net, eq1)).
  5. I don’t think the NO3-N residue in the soil should be considered to “environmental impact”, but the emission should be.

Results and discussion

  1. I suggest to put the 4 figures in figure 3 into one, which will be easier to compare the differences between the treatments.
  2. There was no data support description in line 304-307.
  3. How the data of CO2 emission for AS and CAN come out in Table 5? Think about that, the emission will be different for AS and CAN in field and in where they are.

Conclusions

Recondiser the expression of "did not show significant differences in envrionmental impact".

Round 2

Reviewer 2 Report

The paper has been improved a lot and I think it can be published in your journal.

I still suggest the author putting the 4 figures of Fig.3 into one. This section compared the differences for N release rate of different bio-based fertilizers, putting the 4 figures into one can show clearer differences. As for the authors' worry, I think figure 4 is a good answer.

Regards!
